# ALPHAFOLD DISTILLATION FOR PROTEIN DESIGN

## ABSTRACT

Inverse protein folding, the process of designing sequences that fold into a specific 3D structure, is crucial in bio-engineering and drug discovery. Traditional methods rely on experimentally resolved structures, but these cover only a small fraction of protein sequences. Forward folding models like AlphaFold offer a potential solution by accurately predicting structures from sequences. However, these models are too slow for integration into the optimization loop of inverse folding models during training. To address this, we propose using knowledge distillation on folding model confidence metrics, such as pTM or pLDDT scores, to create a faster and end-to-end differentiable distilled model. This model can then be used as a structure consistency regularizer in training the inverse folding model. Our technique is versatile and can be applied to other design tasks, such as sequence-based protein infilling. Experimental results show that our method outperforms non-regularized baselines, yielding up to 3% improvement in sequence recovery and up to 45% improvement in protein diversity while maintaining structural consistency in generated sequences. Anonymized code for this work is available at `https://anonymous.4open.science/r/AFDistill-28C3`

## 1 INTRODUCTION

Eight of the top ten best-selling drugs are engineered proteins, making inverse protein folding a crucial challenge in bio-engineering and drug discovery (Arnum, 2022). Inverse protein folding involves designing amino acid sequences that fold into a specific 3D structure. Computationally, this task is known as computational protein design and has been traditionally addressed by optimizing amino acid sequences against a physics-based scoring function (Kuhlman et al., 2003). Recently, deep generative models have been introduced to learn the mapping from protein structure to sequences (Jing et al., 2020; Cao et al., 2021; Wu et al., 2021; Karimi et al., 2020; Hsu et al., 2022; Fu & Sun, 2022). While these models often use high amino acid recovery, TM score, and low perplexity as success criteria, they overlook the primary goal of designing novel and *diverse* sequences that fold into the desired structure and exhibit novel functions.

In parallel, recent advancements have also greatly enhanced protein representation learning (Rives et al., 2021; Zhang et al., 2022), structure prediction from sequences (Jumper et al., 2021; Baek et al., 2021b), and conditional protein sequence generation (Das et al., 2021; Anishchenko et al., 2021). While inverse protein folding has traditionally focused on sequences with resolved structures, which represent less than 0.1% of known protein sequences, a recent study improved performance by training on millions of AlphaFold-predicted structures (Hsu et al., 2022). Despite this success, large-scale training is computationally expensive. A more efficient method could be to use a pre-trained forward folding model to guide the training of the inverse folding model.

In this work we construct a framework where the inverse folding model is trained using a loss objective that consists of regular sequence reconstruction loss, augmented with an additional *structure consistency loss (SC)* (see Fig. 1 for system overview). One way of implementing this would be to use folding models, e.g., AlphaFold, to estimate structure from generated sequence, compare it with ground truth and compute TM score to regularize the training. However, a challenge in using Alphafold (or similar) directly is computational cost of inference (see Fig. 2), and the need of ground truth reference structure. Internal confidence structure metrics from folding model can be used instead. However, that approach is still slow for the in-the-loop optimization. To address this, we:

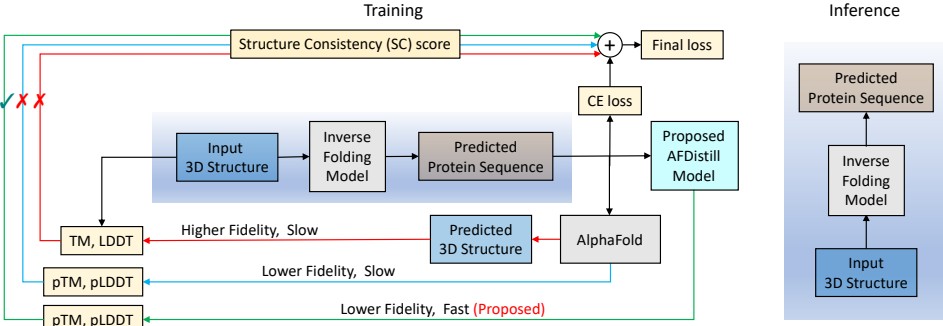

Figure 1: Overview of the proposed AFDistill system. AFDistill contrasts with traditional methods (red line) that use models like AlphaFold to predict protein structure, which is then compared to the actual structure. This method is slow due to model inference times (refer Fig. 2). An alternative (blue line) uses internal metrics from folding model without structure prediction but remains slow and less precise. Our solution, distills AlphaFold's confidence metrics into a faster, differentiable model that offers accuracy akin to AlphaFold, allowing seamless integration into the training process (green line). The improved inverse folding model's inference is shown on the right.

**(i)** Carry out knowledge distillation on AlphaFold and incorporate the resulting model, AFDistill (fixed), into the regularized training of the inverse folding model, which is referred to as structure consistency (SC) loss. The major *novelty* here is that AFDistill enables direct prediction of TM or LDDT scores of a given protein sequence bypassing the structure estimation or the access to ground truth structure. Primary practical benefits of our model include being fast, precise, and end-to-end differentiable. Employing SC loss during training for downstream tasks can be seen as integrating AlphaFold's domain expertise into the model, thereby offering additional boost in its performance.

**(ii)** Perform extensive evaluations, demonstrating that our proposed system surpasses existing benchmarks in structure-guided protein sequence design by achieving lower perplexity, higher amino acid recovery, and maintaining proximity to the original protein structure. Additionally, our system enhances sequence diversity, a key objective in protein design. Due to a trade-off between sequence and structure recovery, our regularized model offers better sequence diversity while maintaining structural integrity. Importantly, our regularization technique is versatile, as evidenced by its successful application in sequence-based protein infilling, where we also observe performance improvement.

**(iii)** Lastly, our SC metric can either be used as regularization for inverse folding, infilling and other protein optimization tasks (e.g., (Moffat et al., 2021)) which would benefit from structural consistency estimation of the designed protein sequence, or as an affordable AlphaFold alternative that provides scoring of a given protein, reflecting its structural content.

## 2 RELATED WORK

**Forward Protein Folding.** Recent computational methods for predicting protein structure from its sequence include AlphaFold (Jumper et al., 2021), which uses multiple sequence alignments (MSAs) and pairwise features. Similarly, RoseTTAFold (Baek et al., 2021a) integrates sequence, 2D distance map, and 3D coordinate information. OpenFold (Ahdritz et al., 2022) replicates AlphaFold in PyTorch. However, due to the unavailability of MSAs for certain proteins and their inference time overhead, MSA-free methods like OmegaFold (Wu et al., 2022), HelixFold (Fang et al., 2022), ESMFold (Lin et al., 2022), and Roney & Ovchinnikov (2022) emerged. These leverage pretrained language models, offering accuracy on par with or exceeding AlphaFold and RoseTTAFold based on the input type.

**Inverse Protein Folding.** Recent algorithms address the inverse protein folding problem of finding amino acid sequences for a specified structure. (Norn et al., 2020) used a deep learning method optimizing via the trRosetta structure prediction network (Yang et al., 2020). (Anand et al., 2022) designed a deep neural network that models side-chain conformers structurally. In contrast, (Jendrusch et al., 2021) employed AlphaFold (Jumper et al., 2021) in an optimization loop for sequence genera-

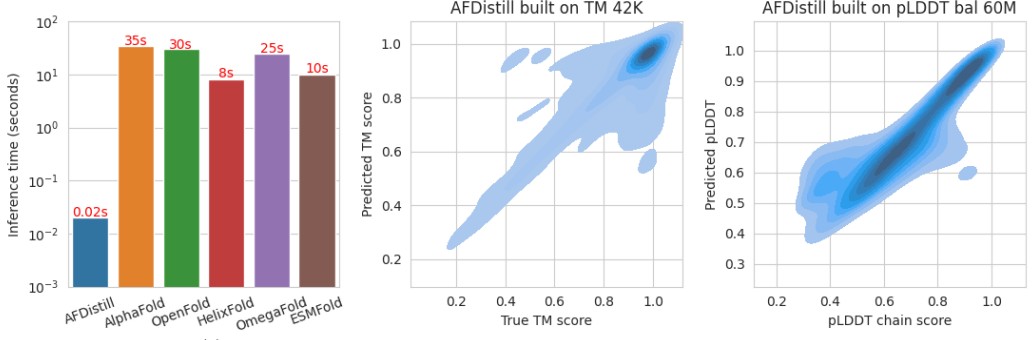

Figure 2: Inference times for protein sequences using our AFDistill model compared to alternatives are displayed on the left. AFDistill maintains fast inference for longer sequences: 0.028s for 1024-length and 0.035s for 2048-length. Timings for AlphaFold and OpenFold (Ahdritz et al., 2022) do not include MSA search times, which can range from minutes to hours. Values for HelixFold (Fang et al., 2022), OmegaFold (Wu et al., 2022), and ESMFold (Lin et al., 2022) are from their publications. The center plot shows kernel density of true vs. AFDistill-predicted TM scores (Pearson's correlation: 0.77), while the right displays a similar plot for pLDDT values (Pearson's correlation: 0.76). Refer to Section 3 for details.

tion, though its use is resource-intensive due to MSA search. MSA-free methods like OmegaFold, HelixFold, and EMSFold are quicker but still slow for optimization loops.

In this work, we propose knowledge distillation from the forward folding algorithm AlphaFold, and build a student model that is small, practical and accurate enough. We show that the distilled model can be efficiently used within the inverse folding model optimization loop and improve quality of designed protein sequences.

## 3  ALPHAFOLD DISTILL

Knowledge distillation (Hinton et al., 2015) transfers knowledge from a large complex model, in our case AlphaFold, to a smaller one, here this is the AFDistill model (see Fig. 3). Traditionally, the distillation would be done using soft labels, which are probabilities from AlphaFold model, and hard labels, the ground truth classes. However, in our case we do not use the probabilities as they are harder to collect or unavailable, but rather the model's predictions (pTM/pLDDT) and the hard labels, TM/LDDT scores, computed based on AlphaFold's predicted 3D structures.

**Scores to Distill**   *TM-score* (Zhang & Skolnick, 2004) measures the mean distance between structurally aligned $C_\alpha$ atoms, scaled by a length-dependent distance parameter, while *LDDT* (Mariani et al., 2013) calculates the average of four fractions using distances between atom pairs based on four tolerance thresholds within a 15Å inclusion radius. Both metrics range from 0 to 1, with higher values indicating more similar structures. *pTM* and *pLDDT* are AlphaFold-predicted metrics for a given protein sequence, representing the model's confidence in the estimated structure. *pLDDT* is a local per-residue score, while *pTM* is a global confidence metric for overall chain reconstruction. In this work, we interpret these metrics as indicators of sequence quality or validity for downstream applications (see Section 4).

### 3.1  DATA

Using Release 3 (January 2022) of AlphaFold Protein Structure Database (Varadi et al., 2021), we collected a set of 907,578 predicted structures. Each of these predicted structures contains 3D coordinates of all the residue atoms as well as the per-resiude pLDDT confidence scores.

To avoid data leakage to the downstream applications, we first filtered out the structures that have 40% sequence similarity or more to the validation and test splits of CATH 4.2 dataset (discussed in

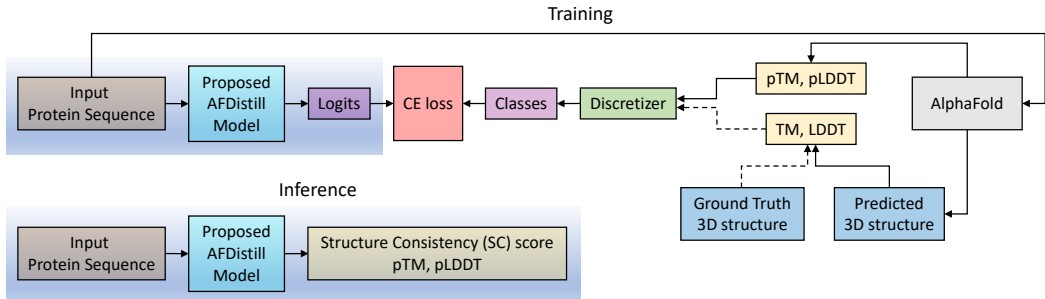

Figure 3: Distillation overview. Top diagram shows the training of AFDistill. The scores from AlphaFold's confidence estimation are denoted as pTM and pLDDT, while the scores which are computed using ground truth and the AlphaFold's predicted 3D structures are denoted as TM and LDDT. These values are then discretized and treated as class labels during cross-entropy (CE) training. Note that the training based on TM/LDTT is limited since the number of known ground truth structures is small. The bottom diagram shows the inference stage of AFDistill, where for each protein sequence it estimates pTM and pLDDT scores.

Table 1: Statistics from January 2022 (left side) and July 2022 (right size) releases of the AlphaFold database. For the earlier release, we created multiple datasets for pTM and pLDDT estimation, while for the later, larger release we curated datasets only for pLDDT estimation.

| Release 3 (January 2022) | | Release 4 (July 2022) | |
|---|---|---|---|
| Name | Size | Name | Size |
| Original | 907,578 | Original | 214,687,406 |
| TM 42K | 42,605 | pLDDT balanced 1M | 1,000,000 |
| TM augmented 86K | 86,811 | pLDDT balanced 10M | 10,000,000 |
| pTM synthetic 1M | 1,244,788 | pLDDT balanced 60M | 66,736,124 |
| LDDT 42K | 42,605 | | |
| pLDDT 1M | 905,850 | | |

Section 4). Then, using the remaining structures, we created our pLDDT 1M dataset (see Table 1), where each protein sequence is paired with the sequence of per-residue pLDDTs. Additionally, to reduce the computational complexity of AFDistill training, we limited the maximum protein length to 500 by randomly cropping a subsequence.

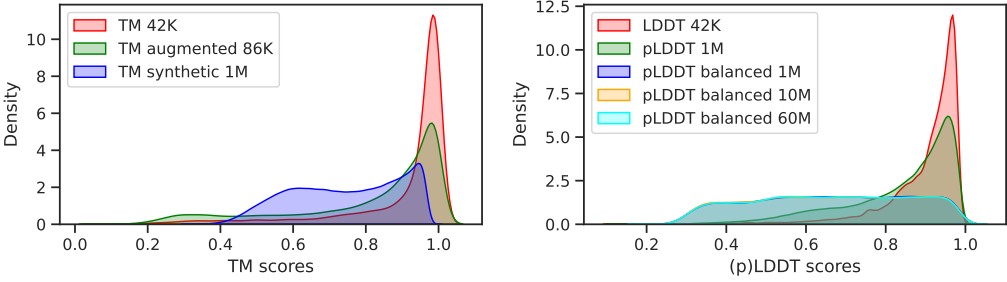

Figure 4: Distribution of the (p)TM/(p)LDDT scores in various datasets used in AFDistill training.

We created datasets based on true TM and LDDT values using predicted AlphaFold structures. Specifically, using the PDB-to-UniProt mapping, we selected a subset of samples with matching ground truth PDB sequences and 3D structures, resulting in 42,605 structures. We denote these datasets as TM 42K and LDDT 42K (see Table 1). Fig. 4 shows their score density distribution, which is skewed towards higher values. To address data imbalance, we curated two additional TM-based datasets. TM augmented 86K was obtained by augmenting TM 42K with a set of perturbed original

Table 2: Validation loss of AFDistill on datasets from Table 1 (For more details, see Tables 5 and 6.)

| Training data | Val CE loss | Training data | Val CE loss |
|---|---|---|---|
| TM 42K | **1.10** | LDDT 42K | 3.39 |
| TM augmented 86K | 2.12 | pLDDT 1M | 3.24 |
| pTM synthetic 1M | 2.55 | pLDDT balanced 1M | 2.63 |
| | | pLDDT balanced 10M | 2.43 |
| | | pLDDT balanced 60M | **2.40** |

protein sequences (permuted/replaced parts of protein sequence), estimating their structures with AlphaFold, computing corresponding TM-score, and keeping the low and medium range TM values. pTM synthetic 1M was obtained by generating random synthetic protein sequences and feeding them to AFDistill (pre-trained on TM 42K data), to generate additional data samples and collect lower-range pTM values. The distribution of the scores for these additional datasets is also shown in Fig. 4, where both TM augmented 86K and pTM synthetic 1M datasets are less skewed, covering lower (p)TM values better.

Using Release 4 (July 2022) with over 214M predicted structures, we observed a similar high skewness in pLDDT values. To mitigate this, we filtered out samples with upper-range mean-pLDDT values, resulting in a 60M sequences dataset, with additional 10M and 1M versions created. Their density is shown in Fig. 4.

In summary, AFDistill is trained to predict both the actual structural measures (TM, LDDT, computed using true and AlphaFold's predicted structures) as well as AlphaFold's estimated scores (pTM and pLDDT). In either case the estimated structural consistency (SC) score is well correlated with its target (refer to Fig.2) and can be used as an indicators of protein sequence quality or validity.

## 3.2 MODEL

AFDistill model is based on ProtBert (Elnaggar et al., 2020), a Transformer BERT model (420M parameters) pretrained on a large corpus of protein sequences using masked language modeling. For our task we modify ProtBert head by setting the vocabulary size to 50 (bins), corresponding to discretizing pTM/pLDDT in range (0,1). For pTM (scalar) the output corresponds to the first ⟨CLS⟩ token of the output sequence, while for pLDDT (sequence) the predictions are made for each residue position.

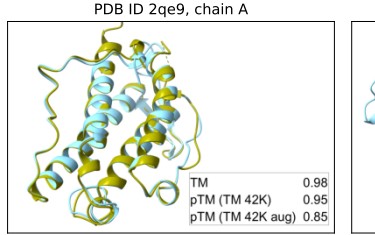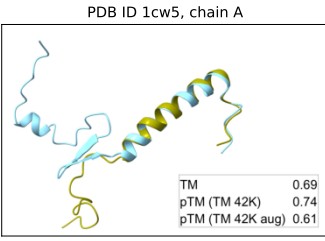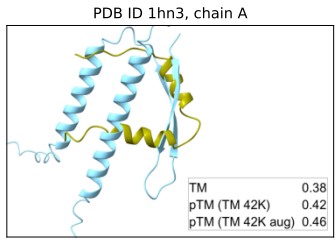

Figure 5: Examples of 3D protein structures from the dataset, corresponding to high, medium, and low actual TM scores (top row in legend), as well as AFDistill predictions, trained on TM 42K (middle row) and TM augmented 86K (bottom row).

## 3.3 DISTILLATION EXPERIMENTAL RESULTS

In this section, we discuss the model evaluation results after training on the presented datasets. To address data imbalance, we used weighted sampling during minibatch generation and Focal loss (Lin et al., 2017) instead of traditional cross-entropy loss. Table 2 shows results for (p)TM-based datasets. AFDistill trained on TM 42K performed the best, followed by augmented and synthetic data. For (p)LDDT-based datasets, increasing scale and data balancing improved validation performance.

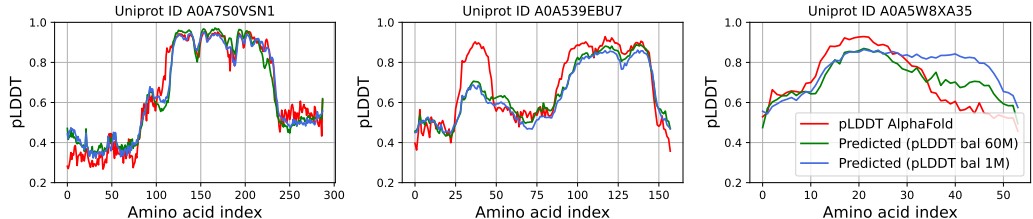

Figure 6: Dataset examples of the per-residue predictions for two AFDistill models (blue and green lines), build on pLDDT balanced 1M and 60M datasets, versus the AlphaFold predictions (red line).

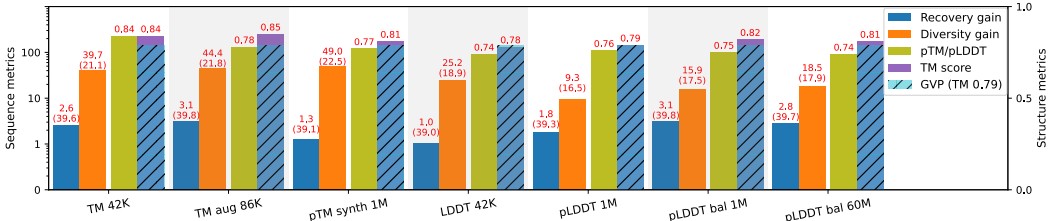

Figure 7: Evaluation results for GVP with SC regularization are shown with various AFDistill pretraining datasets on the x-axis. The left y-axis displays sequence metrics (recovery and diversity gains), and the right y-axis shows structure metrics (TM and SC scores). Blue and orange solid bars represent recovery and diversity gains over the vanilla GVP baseline (38.6 in recovery, 15.1 in diversity, and 0.79 in TM score). Olive and purple bars display predicted SC and test set TM scores, respectively, while dashed cyan bar shows the baseline GVP TM score. TM 42K and TM augmented 86K pretrained AFDistill models achieve the best overall performance, with high diversity and moderate improvement in sequence and structure recovery.

In Fig. 2 we show kernel density plots of the true vs pTM scores and pLDDT values on the entire validation set. Majority of the density is concentrated along the diagonal, indicating that the predicted scores match well with the ground truth. The mismatches are grouped in off-diagonal regions, but these areas have low density, indicating that the predictions are still accurate. Moreover, since the data for TM 42K is skewed towards 1.0 (top density plot in Fig. 4), most of the data is clustered in upper left corner. On the other hand, for the dataset pLDDT bal 60M which is balanced (bottom panel in Fig. 4), the predictions and true values are spread more uniformly along the diagonal.

Finally, in Fig. 5 and 6, we show a few examples of the data samples together with the corresponding AFDistill predictions. Fig. 2 and Fig. 12, 13 (in Appendix) also show plots of SC (pTM or pLDDT) versus TM score, indicating that AFDistill is a viable tool for regularizing protein representation to enforce structural consistency or structural scoring of protein sequences, reflecting its overall composition and naturalness (in terms of plausible folded structure).

## 4 INVERSE PROTEIN FOLDING DESIGN

In this section we demonstrate the benefit of applying AFDistill as a structure consistency (SC) score for solving the task of inverse protein folding, as well as for the protein sequence infilling as a means to novel antibody generation. The overall framework is presented in Fig. 1 (following the green line in the diagram), where the traditional inverse folding model is regularized by our SC score. Specifically, during training, the generated protein is fed into AFDistill, for which it predicts pTM or pLDDT score, and combined with the original CE training objective results in

$$\mathcal{L} = \mathcal{L}_{\text{CE}} + \alpha \mathcal{L}_{\text{SC}}, \tag{1}$$

where $\mathcal{L}_{\text{CE}} = \sum_1^N \mathcal{L}_{\text{CE}}(\mathbf{s}_i, \hat{\mathbf{s}}_i)$ is the CE loss, $\mathbf{s}_i$ is the ground truth and $\hat{\mathbf{s}}_i$ is the generated protein sequence, $\mathcal{L}_{\text{SC}} = \sum_{i=1}^N (1 - SC(\hat{\mathbf{s}}_i))$ is the structure consistency loss, $N$ the number of training sequences, and $\alpha$ is the weighting scalar for the SC loss, in our experiment it is set to 1.

**Metrics** We use standard sequence evaluation metrics to measure prediction design quality. *Recovery* (0-100) is the normalized average of exact matches between predicted and ground truth sequences. *Diversity* (0-100) is the complement of the average recovery for pairwise comparisons in a set. We aim for high recovery and high diversity rates. *Perplexity* measures sequence likelihood, with lower values indicating better performance. For structure evaluation, we use *TM-score* and *structure consistency (SC)* score, which is AFDistill's output for a given input.

## 4.1 RESULTS

We present experimental results for several recently proposed deep generative models for protein sequence design accounting for 3D structural constraints. For the inverse folding tasks we use CATH 4.2 dataset, curated by (Ingraham et al., 2019). The training, validation, and test sets have 18204, 608, and 1120 structures, respectively. While for protein infilling we used SabDab (Dunbar et al., 2013) dataset curated by (Jin et al., 2021) and focus on infilling CDR-H3 loop. The dataset has 3896 training, 403 validation and 437 test sequences.

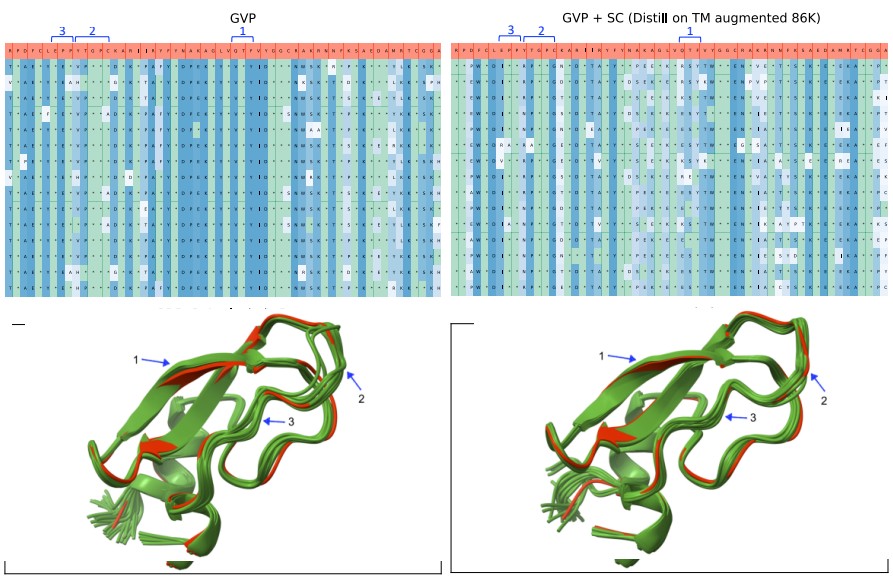

Figure 8: The comparison between baseline GVP (left) and SC regularized GVP (right) using AFDistill pre-trained on TM augmented 86K dataset shows 15 generated protein sequences from each model. Green cells with * indicate amino acid identity with ground truth (top red row), while blue cells represent novelty. The shade of blue indicates amino acid frequency in the column (darker = more frequent, lighter = rare). High recovery and diversity rates are seen with many green and light blue cells. Bottom plots display AlphaFold estimated structures (green) and ground truth (red). Recovery is 40.8 and diversity is 11.2 for GVP, while for GVP+SC, it is 42.8 and 22.6, respectively. SC-regularized GVP has accurate reconstructions with high sequence diversity, while GVP alone exhibits more inconsistencies, marked with arrows.

**GVP** Geometric Vector Perceptron GNNs (GVP) (Jing et al., 2020) is the inverse folding model, that for a given target backbone structure, represented as a graph over the residues, replaces dense layers in a GNN by simpler layers, called GVP layers, directly leveraging both scalar and geometric features. This allows for the embedding of geometric information at nodes and edges without reducing such information to scalars that may not fully capture complex geometry. The results of augmenting GVP training with SC score regularization are shown in Fig. 7 (see also Appendix, Table 13 for additional results).

Baseline GVP without regularization achieves 38.6 in recovery, 15.1 in diversity, and 0.79 in TM score on the test set. Employing SC regularization leads to consistent improvements in sequence recovery (1-3%) and significant diversity gain (up to 45%) while maintaining high TM scores. pTM-based SC scores show a better overall influence on performance compared to pLDDT-based ones. It's important

| | Recovery | Diversity |
|---|---|---|
| Core | 52.7 | 9.8 |
| Surface | 32.6 | 23.1 |
| All | 39.6 | 21.1 |

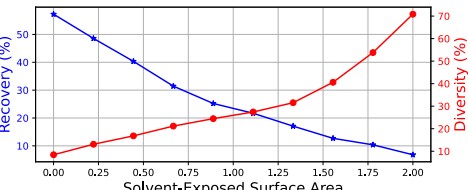

Figure 9: Distribution of recovery and diversity values across residues in generated structures. Left Table shows the analsys with respect to core ($\geq$ 24 neighbors), surface ($\leq$ 16 neighbors) and all residue categories. Right Figure shows similar analysis with respect to solvent-exposed surface area.

to note that AFDistill's validation performance on distillation data doesn't always reflect downstream application performance. For example, TM augmented 86K outperforms TM 42K, despite having slightly worse validation CE loss. This suggests that augmented models may enable more generalized sequence-structure learning and provide a greater performance boost for inverse folding models.

Fig. 8 further demonstrates the effect of recovery and diversity on protein sequences and AlphaFold-generated 3D structures for GVP and GVP+SC models. GVP+SC achieves higher diversity while retaining accurate structure. Recovery and diversity scores are 40.8 and 11.2 for GVP, and 42.8 and 22.6 for GVP+SC, respectively. The bottom plots display AlphaFold estimated structures (green) and ground truth (red). Despite its high sequence diversity, GVP+SC shows very accurate reconstructions (average TM score 0.95), while GVP exhibits more inconsistencies (TM score 0.92), marked with blue arrows.

Table 3: Evaluation results of ProteinMPNN trained with and without SC regularization (AFDistill trained on TM aug 86K dataset). The values in parenthesis show gain on test set of using SC-regularized training as compared to original training.

| | Recovery | | Diversity | | Perplexity | |
|---|---|---|---|---|---|---|
| | ProteinMPNN | ProteinMPNN +SC | ProteinMPNN | ProteinMPNN +SC | ProteinMPNN | ProteinMPNN +SC |
| Backbone Noise 0.02 | 47.7 | 47.5 (-0.4%) | 22.5 | 24.3 (+8.0%) | 5.1 | 5.1 (+0.0%) |
| Backbone Noise 0.1 | 43.8 | 44.0 (+0.5%) | 28.1 | 30.4 (+8.2%) | 5.3 | 5.4 (+1.9%) |
| Backbone Noise 0.2 | 39.5 | 39.9 (+1.0%) | 31.3 | 34.4 (+9.9%) | 5.8 | 5.8 (+0.0%) |
| Backbone Noise 0.3 | 36.3 | 36.4 (+0.0%) | 33.0 | 37.8 (+14.6%) | 6.2 | 6.3 (+1.6%) |

We also conducted a core/surface analysis similar to Hsu et al. (2022) to examine recovery and diversity distribution across residues in generated structures. Residues were categorized by the density of neighboring $C_\alpha$ atoms within 10A (core: $\geq$ 24 neighbors; surface: $\leq$ 16 neighbors). Using SC-regularized GVP results on TM 42K dataset, we computed recovery and diversity scores per class (Fig. 9). Core residues, being more constrained, better match ground truth with less diversity. Surface residues, being less constrained, exhibit lower recovery but higher diversity as the model has more freedom in selecting residues.

Additionally, we plot the solvent-exposed surface area sas versus recovery and diversity, computed per residue (see right panel in Fig. 9). As expected, the recovery plot shows negative correlation and diversity plot shows positive correlation as the surface area increases. Solvent-exposed surface area was calculated using GROMACS software suite gro with default parameters.

**Note on sequence diversity** In Section G of Appendix we offer a set of experiments to shed some light on why SC regularization leads to improved sequence diversity. In particular in Fig.14 we show that the main source of diversity is in the limited guidance from AFDistill about the specific sequence to generate to match a given 3D structure, since it does not have access to the structural information, allowing many relevant sequences with high pTM/pLDDT to be considered as good candidates. AFDistill regularization during training injects candidate sequences which have high pTM/pLDDT scores, therefore likely matching the input structure better, thus ensuring high recovery rate. At the same time these sequences differ from the ground truth, thus promoting diversity (see Section G for more details).

Table 4: Experiments on PiFold comparing the performance metrics on the test set of CATH 4.2 for different model variants (original vs SC-regularized training based on different AFDistill models) using greedy and sampled decoding strategies. The values in parentheses represent the percentage change with respect to the original PiFold model.

| | Original | | TM 42K | | TM aug 86K | | TM synth 1M | | LDDT 42K | | pLDDT 1M | | pLDDT bal 60M | |
|---|---|---|---|---|---|---|---|---|---|---|---|---|---|---|
| | Rec | Perp | Rec | Perp | Rec | Perp | Rec | Perp | Rec | Perp | Rec | Perp | Rec | Perp |
| Greedy | 51.1 | 4.8 | 50.9 (-0.4%) | 5.0 (+4.0%) | 51.0 (-0.2%) | 4.8 (+0.0%) | 50.5 (-1.2%) | 5.2 (+8.3%) | 50.8 (-0.6%) | 4.9 (+2.1%) | 50.9 (-0.4%) | 4.8 (+0.0%) | 51.1 (+0.0%) | 4.7 (-2.1%) |
| | Rec | Div | Rec | Div | Rec | Div | Rec | Div | Rec | Div | Rec | Div | Rec | Div |
| Sampled | 42.6 | 52.4 | 42.5 (-0.2%) | 60.7 (+15.8%) | 42.8 (+0.5%) | 60.2 (+14.9%) | 42.4 (-0.5%) | 61.1 (+16.6%) | 42.3 (-0.7%) | 60.9 (+16.2%) | 42.5 (-0.2%) | 60.5 (+15.5%) | 42.9 (+0.7%) | 60.0 (+14.5%) |

**ProteinMPNN**  ProteinMPNN model Dauparas et al. (2022) is a recent protein design model, which is based on message passing neural network (MPNN) with specific modifications to improve amino acid sequences recovery of proteins given their backbone structures. The model incorporates structure features, edge updates, and an autoregressive approach for decoding the sequences. In Table 3 we compared the results of original unmodified training of ProteinMPNN to the SC-regulared training (AFDistill model trained on TM aug 86K dataset). We also varied ProteinMPNN internal parameter, which adds noise to the input backbone protein structure. As can be seen, SC regularization maintains high recovery and perplexity rates while improving the diversity of the generated protein sequences. Backbone noise, which is a part of ProteinMPNN model, can also be seen as a form of regularization, however while the increase in noise leads to improved sequence diversity it also leads to the decrease in amino acid recovery rate. SC regularization, on the other hand, promotes diverse generation and maintains high sequence recovery rates.

**PiFold**  PiFold Gao et al. (2023) is another recent protein design model which introduces a new residue featurizer and stacked PiGNNs (protein inverse folding graph neural networks). The residue featurizer constructs residue features and introduces learnable virtual atoms to capture information that could be missed by real atoms. The PiGNN layer learns representations from multi-scale residue interactions by considering feature dependencies at the node, edge, and global levels. In Table 4 we present the results of original and SC-regularized PiFold (using different AFDistill models). We note that the original PiFold evaluation was based on using greedy decoding to generate a sequence. Following the standard practice (GVP, GraphTrans, ProteinMPNN, etc), we have included also the results based on sampling (using 100 samples per sequence) to match other works and compute sequence diversity score. The results show that SC regularization based on AFDistill trained on TM aug 86K results in a near-identical recovery rate compared to the original model, while notably enhancing sequence diversity. This indicates an improvement in PiFold's performance by maintaining recovery rates while increasing the variety of generated sequences. Also observe a decrease in recovery rates for sampled generation as compared to greedy decoding across all the models.

**Additional Experiments**  In Section H of Appendix we present additional experiments using ESM-IF Hsu et al. (2022) and Graph Transformer (Wu et al., 2021). Finally, in addition to protein design, we also examine the effect of SC regularization on a protein infilling task to design complementarity-determining regions (CDRs) of an antibody sequence.

## 5 CONCLUSION

In this work we introduce AFDistill, a distillation model based on AlphaFold, which for a given protein sequence estimates its structural consistency (SC: pTM or pLDDT) score. We provide experimental results to showcase the efficiency and efficacy of the AFDistill model in high-quality protein sequence design, when used together with many of the current state of the art protein inverse folding models or large protein language model for sequence infilling. Our AFDistill model is fast and accurate enough so that it can be efficiently used for regularizing structural consistency in protein optimization tasks, maintaining sequence and structural integrity, while introducing diversity and variability in the generated proteins.

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

SUPPLEMENTARY MATERIAL FOR PAPER SUBMISSION
ALPHAFOLD DISTILLATION FOR INVERSE PROTEIN FOLDING

TABLE OF CONTENTS

## A    LIMITATIONS OF THE PROPOSED WORK

Although our proposed AFDistill system is novel, efficient and showed promising results during evaluations, there are a number of limitations of the current approach:

- AFDistill dependency on the accuracy of the AlphaFold forward folding model: The quality of the distilled model is directly related to the accuracy of the original forward folding model, including the biases inherited from it.

- Limited coverage of protein sequence space: Despite the advances in AlphaFold forward folding models, they are still limited in their ability to accurately predict the structure of many protein sequences, including the TM score and pLDDT confidence metrics, that AFDistill relies on.

- Uncertainty in structural predictions: The confidence metrics (TM score and pLDDT) used in the distillation process are subject to uncertainty, which can lead to errors in the distilled model's predictions and ultimately impact the quality of the generated sequences in downstream applications.

- The need for a large amount of computational resources: The training process of AFDistill model requires significant computational resources. However, this might be mitigated by the amortization effect where the high upfront training cost in downstream applications pays in terms of cheap and fast inference through the model.

## B    BACKGROUND ON PROTEIN DESIGN

A protein is a linear chain of variable length made up of twenty amino acids, also called residues. These are denoted by 20 characters (A-Alanine, G-Glycine, I-Isoleucine, L-Leucine, P-Proline, V-Valine, F-Phenylalanine, W-Tryphtophan, Y-Tyrosine, D-Aspartic Acid, E-Glutamic Acid, R-Arginine, H-Histidine, K-Lysine, S-Serine, T-Threonine, C-Cystene, M-Methionine, N-Asparagine, Q-Glutamine). Each amino acid has the same core structure (backbone), consisting of alpha carbon atom $C_\alpha$, connected to an amino group $NH2$, carboxyl group $COOH$ and hydrogen atom $H$. The backbone is identical in all amino acids, while the variable group, called side chain, which is also attached to alpha carbon $C_\alpha$, is always different and determines the amino acid, including its chemical and mechanical properties. Amino acids are attached to each other by a covalent bond, known as peptide bond (carboxyl group $COOH$ of one amino acid and the amino group $NH2$ of the other amino acid combine, releasing water molecule $H2O$ and create a peptide bond). In this work, as is commonly done, we define protein 3D structure specified only by the $C_\alpha$ atoms of amino acids.

The protein inverse folding task is to draw a sequence from the true distribution of $n$-length sequences of amino acids $Y \in \{1, \ldots, 20\}$, conditioned on a fixed protein structure, such that the designed protein folds into that structure. The protein structure can be represented as an attributed graph $G = (V, E)$ with node features $V = \{v_1, \ldots, v_N\}$, describing each residue and edge features $E = \{e_{ij}\}$, capturing relationships between them. Thus, the final conditional distribution we are interested in modeling is: $P(Y|X) = p(y_i, \ldots, y_n|X)$, which is known as computational protein design task.

Protein structures are intrinsically dynamic and each structure thus possess high designability, i.e. the total number of amino acid sequences that can fold to a target protein structure is high, without losing stability of the structure. The highly designable structures always enjoy beneficial properties such as higher thermodynamic stability, mutational stability, fast folding, functional robustness, etc. Therefore, we need to learn a "soft" function that can model this high designability associated with a protein structure, i.e. generating diverse sequences for a given protein structure.

## C    ALPHAFOLD MODEL OVERVIEW

A schematic overview of AlphaFold model is shown in Fig. 10, which it takes as input a protein sequence and produces as output, among others, the predicted 3D structure, as well as the confidence estimates of its prediction, pTM and pLDDT, which measure the estimated confidence of how well the predicted and ground truth structures match.

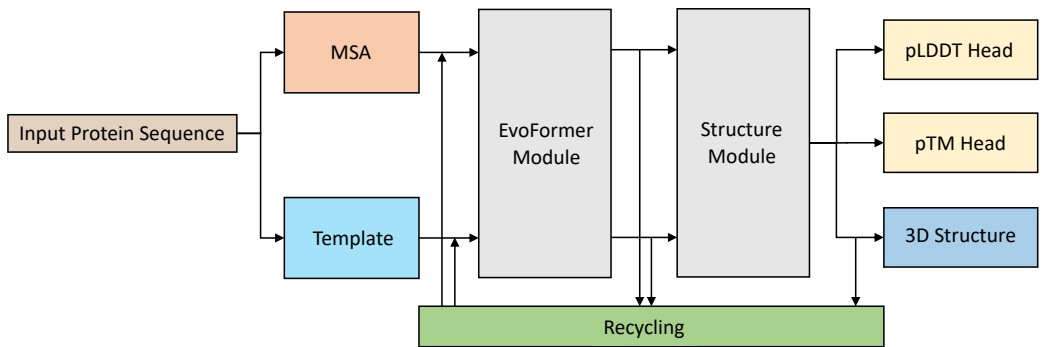

Figure 10: Overview of the inference stage in AlphaFold model. Given an input protein sequence, first the search is performed in genetic database to find similar sequences and construct multiple sequence alignments (MSA). Then a structure database search is done to find similar 3D structures and construct templates. The MSA and templates are fed into EvoFormer module, whose output is then sent to the Structure module, which is finally completed with the multiple output heads. The 3D structure head generates predicted 3D protein structure, while pLDDT and pTM heads estimate the confidence of the computed structure. Optionally, the generated structure together with the intermediate states are recycled and sent back to update/correct MSA and template representations for further processing and improvement.

## D    AFDISTILL TRAINING

Tables 5, 6 show the validation performance of AFDistill trained on each of the (p)TM-based and (p)LDDT-based datasets, respectively. Table 7 shows results on (p)LDDT chain-based datasets. Note that (p)LDDT chain is the dataset, similar to (p)TM datasets, where for each sequence we associate a single scalar, in this case the average of all the per-reside (p)LDDT values.

## E    AFDISTILL SCATTER PLOTS OF PREDICTIONS

In Fig. 11 we show scatter plots of the true vs pTM scores and pLDDT values on the entire validation set. We see a clear diagonal pattern in both plots, where the predicted and true values match. There are also some number of incorrect predictions (reflected along the off-diagonal), where we see that for the true scores in the upper range, the predicted scores are lower, indicating that AFDistill tends to underestimate them.

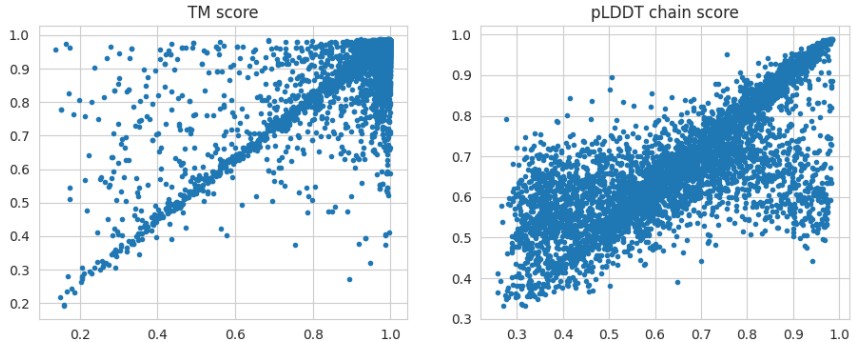

Figure 11: A scatter plot of the true and predicted TM score (left panel, data: TM 42K) and pLDDT (right panel, data: pLDDT bal 60M) for the data presented in main text in Fig. 2.

Table 5: Validation CE loss for the AFDistill model trained on each of the (p)TM-based datasets. To address data imbalance during training, we employed weighted sampling for minibatch generation to so that the TM-scores cover their range (0,1) close to uniform distribution. Moreover, we also used Focal loss (Lin et al., 2017) in place of the standard cross-entropy (CE) loss (the evaluation is still done using CE loss across all the training setups). Based on the validation loss, we see that the AFDistill model trained on TM 42K dataset performed the best, followed by the dataset with augmentations, and the synthetic performed the worst. We also see that weighted sampling and focal loss do help in addressing the data imbalance problem, although for TM augmented 86K, the balanced augmentation seemed to help better and the best performance was for the case when no weighted sampling is applied and the traditional CE loss is used. As shown in Section 4, the validation performance on the distillation data may not always indicate the performance on the downstream applications, where in particular we observed that the Distill model, trained on TM augmented 86K dataset, overall performed better than TM 42K, while having slightly worse validation CE loss.

| Data | Training | | Validation |
|---|---|---|---|
| | Weighted sampling | Focal loss ($\gamma$) | CE loss |
| TM 42K | − | − | 1.33 |
| | + | − | 1.37 |
| | + | 1.0 | 1.16 |
| | + | 3.0 | **1.10** |
| | + | 10.0 | 1.29 |
| TM augmented 86K | − | − | **2.12** |
| | + | 1.0 | 2.15 |
| | + | 3.0 | 2.19 |
| | + | 10.0 | 2.25 |
| pTM synthetic 1M | − | − | 2.90 |
| | + | 1.0 | 2.75 |
| | + | 3.0 | **2.55** |
| | + | 10.0 | 3.20 |

Table 6: Validation CE loss for AFDistill trained on each of the (p)LDDT-based datasets. We see that weighted sampling coupled with Focal loss, performed the best.

| Data | Training | | Validation |
|---|---|---|---|
| | Weighted sampling | Focal loss ($\gamma$) | CE loss |
| LDDT 42K | - | - | 3.47 |
| | + | 1.0 | 3.44 |
| | + | 3.0 | 3.42 |
| | + | 10.0 | **3.39** |
| pLDDT 1M | - | - | 3.27 |
| | + | 1.0 | 3.28 |
| | + | 3.0 | **3.25** |
| | + | 10.0 | 3.24 |

## F ARCHITECTURAL AND TRAINING DETAILS

### F.1 AFDISTILL

Table 8 shows architectural details of AFDistill and ProtBert, while in Table 9 we present training details for AFDistill for two experimental setups. In all the experiments we used A100 GPUs. From the tables we can see that the AFDistill (420 M parameters) training takes approximately 24 hours on 1 GPU for TM 42K dataset, and 7 days on 8 GPUs for pLDDT balanced 60M dataset. Note

Table 7: Validation CE loss for the AFDistill model trained on each of the (p)LDDT chain-based datasets. (p)LDDT chain is the dataset, similar to (p)TM datasets, where for each sequence we associate a single scalar, in this case the average of all the per-reside (p)LDDT values. Similar as before, we see that the use of weighted sampling coupled with Focal loss helps in boosting the model performance. We also see that increasing the scale of data (which is already balanced) improves the performance even further.

| Data | Training | | Validation |
|---|---|---|---|
| | Weighted Sampling | Focal loss ($\gamma$) | CE loss |
| LDDT chain 42K | - | - | 3.69 |
| | + | 1.0 | 3.57 |
| | + | 3.0 | 3.63 |
| | + | 10.0 | **3.59** |
| pLDDT chain 1M | - | - | 3.29 |
| | + | 1.0 | 3.36 |
| | + | 3.0 | **3.30** |
| | + | 10.0 | 3.31 |
| pLDDT chain balanced 1M | – | – | 2.45 |
| pLDDT chain balanced 10M | – | – | 2.24 |
| pLDDT chain balanced 60M | – | – | **2.21** |

that AFDistill training cost is amortized: the model is trained once and reused it many downstream applications. During training of the downstream applications, AFDistill model needs to be kept in memory to compute SC (structural consistency) score.

Table 8: Architectural details of AFDistill and ProtBert (which is used to initialize AFDistill training).

| Model | Number of parameters | Number of layers | Hidden layer size | Number of heads | Vocab size | Pretraining Data | Reference |
|---|---|---|---|---|---|---|---|
| ProtBert | 420M | 30 | 1024 | 16 | 30 (20 amino acids + 10 aux tokens) | BFD100 (572 GB, 2B proteins) Uniref100 (150 GB, 216M proteins) | (Devlin et al., 2018) |
| AFDistill | 420M | 30 | 1024 | 16 | 50 (50 bins, TM/pLDDT (0,1)) | TM 42K (20 MB, 42K sequences) pLDDT balanced 60M (100 GB, 60M sequences) | – |

Table 9: Training details for AFDistill for two experimental setups: small - using TM 42K dataset, and large - using AFDistill pLDDT balanced 60M.

| Model | Learning rate | Batch size | Optimizer | GPUs | Training time |
|---|---|---|---|---|---|
| AFDistill TM 42K | $1e^{-6}$ | 10 | Adam | $1 \times$ A100, 40GB | 1 day |
| AFDistill pLDDT bal 60M | $1e^{-6}$ | 10 | Adam | $8 \times$ A100, 40GB | 7 days |

## F.2 PROTEIN DESIGN

Table 10 shows training details for GVP and ProteinMPNN models. Additionally, we note that for the original PiFold model it takes 60 epochs (6 hours on 1 GPU) to train the model, while for PiFold+SC it takes 60 epochs (8 hours on 1 GPU) to do the training. The increased training time is due to frequent validations (which involves sampling 100 samples per sequence for recovery and diversity computations). Note, that once the downstream application is trained, AFDistill is not used during inference.

Table 10: Training details for GVP and ProteinMPNN (original, as well as SC-regularized using our AFDistill model). Note that although AFDistill has 420M parameters, these are not part of the learnable model parameters, therefore are not counted towards the total.

| Setup | Parameters | Learning rate | Batch size | Optimizer | GPUs | Training time |
|---|---|---|---|---|---|---|
| GVP / GVP+SC | 1M | $1e^{-3}$ | 3000 res/batch | Adam | $1 \times$ A100, 40GB | 1 day |
| ProteinMPNN / ProteinMPNN + SC | 1.6M | varied | 5000 res/batch | Adam | $1 \times$ A100, 40GB | 2 days |

## G  GVP TRAINING DETAILS

An example of GVP training progress regularized by the structure consistency (SC) score computed by the AFDistill model (pre-trained on various (p)TM-based datasets) is shown in Fig. 12. This figure shows that although SC score may be less accurate on the absolute scale, on the relative scale we can see it accurately detecting decays and improvements in the sequence quality as the GVP trains. Similarly, in Fig. 13 we show scatter plots of estimated pTM versus true TM scorefor GVP-generated protein sequences regularized by SC score.

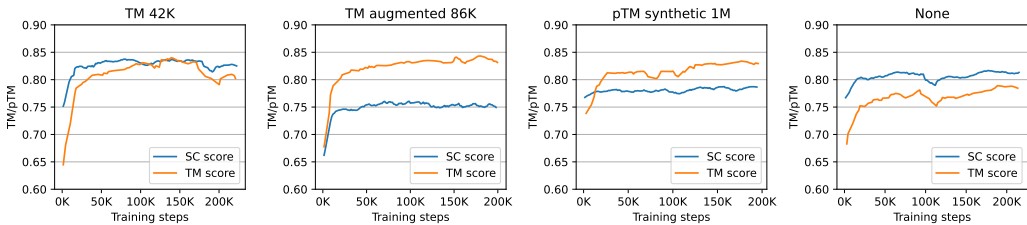

Figure 12: Example of the training progress (on CATH 4.2 dataset) of the GVP model regularized by the structure consistency (SC) score computed by the AFDistill model pre-trained on different datasets. Each plot shows the results for one of the Distill pre-training datasets, where the blue line represents the SC score computed by the AFDistill model (in this case generating pTM value), while the orange line shows the actual TM score computed between the ground truth structure and the AlphaFold's estimated 3D structures for a GVP-generated protein sequences. The last plot on the right shows the original, unregularized GVP training, where SC score was computed but never applied as part of the loss. It can be seen that SC correlates well with the TM score for TM 42K, while for others (TM augmented 86K and pTM synthetic 1M datasets) it tends to underestimate true TM score. Therefore, SC score may be less accurate on the absolute scale, while on the relative scale we can see that it can accurately detect decays and improvements in the sequence quality as the GVP trains. And the latter is of particular importance for SC to be a regularization loss during training, since it can clearly identify the ill-generated protein sequences early in the training (lower SC scores) and recognize well-defined sequences later during the training (higher SC scores).

### G.1  EFFECT OF USING AFDISTILL TRAINED FROM SCRATCH

We also experimented with AFDistill models trained from scratch (as opposed to starting from pre-trained ProtBert), but observed worse performance. As an example, we trained AFDistill from scratch on TM42K dataset. The validation CE loss during distillation was 1.5 (versus 1.1 when using pre-trained ProtBert model). Moreover, training of AFDistill model from scratch takes longer (3 days vs 1 day). When regularizing GVP with AFDistill from scratch, we get similar recovery rate (39.4 vs 39.6) but lower sequence diversity (15.9 vs 21.1), which confirms the benefit of common practice of fine-tuning the pretrained models as opposed to starting from random models weights.

### G.2  EFFECT OF STRUCTURE CONSISTENCY (SC) SCORE ON GVP PERFORMANCE

For protein design (e.g., using GVP as a base model) the objective is CE + SC (cross-entropy + AFDistill structure consistency score). In Fig. 14 we present the effect of SC magnitude on the GVP performance on the test set of CATH dataset. As can be seen, when only the CE term is present (the blue left most bar in both panels, representing the original GVP), the model is encouraged to

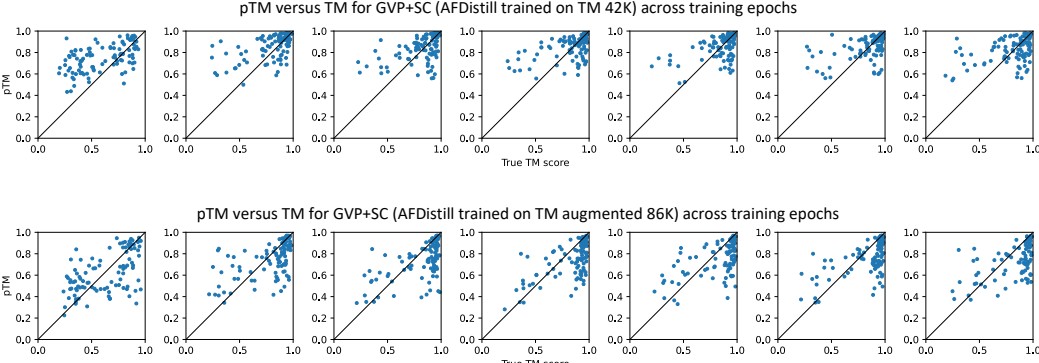

Figure 13: Estimated pTM versus true TM score (based on AlphaFold structure prediction) for GVP-generated protein sequences regularized by SC score. The top row shows results for SC computed by AFDistill model trained on TM 42K, while the bottom row is for AFDistill trained on TM augmented 86K. The columns in each row correspond to the progress as GVP trains. Note that the top row corresponds to the first left plot in Fig. 12, while the bottom row corresponds to the second plot in Fig. 12. It can be observed that in the earlier stages of GVP training, the generated protein sequences are of poor quality, reflected in pTM and TM scores that are spread across the (0,1) range. On the other hand, as the training progresses, the generated sequences are getting better and the pTM/TM score is concentrated more in the upper range. Another observation is that for AFDistill trained on TM 42K dataset, the predicted and true TM score are better aligned across the diagonal (compare with orange and blue lines on the left plot in Fig. 12), while for AFDistill trained on TM augmented 86K dataset, pTM tends to underestimate true TM score. These plots show that AFDistill is viable sequence scoring tool, which fairly accurately measures the structural consistency of the generated protein sequences. Combined with the fact that it is fast and end-to-end differentiable, shows its potential for many of the protein optimization problems.

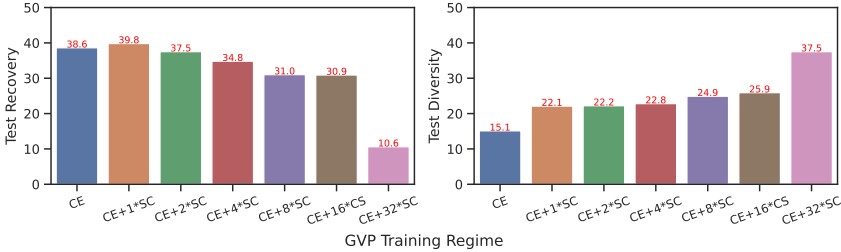

Figure 14: The effect of Structure Consistency (SC) loss on the performance of GVP. Left panel shows the amino acid recovery rate and the right panel shows the diveristy rate on the test set of CATH dataset. The horizontal y-axis shows the different choices of objective function during training: CE is the cross-entropy loss, SC is the Structure Consistency score computed by AFDistill.

recover the specific ground truth protein sequence for a given 3D structure, and this promotes model accuracy, and high amino acid recovery rate, while also resulting in low diversity. On the other hand, when only the SC term is present (the pink right most bar, reprenting CE+32*SC, i.e., when SC completely dominates and CE can be ignored), this results in poor and degenerated protein sequences. This is expected, since AFDistill alone cannot guide GVP which sequence it should generate to match the given input 3D structure. Recall, that AFDistill has no information about the structure, and since many of the relevant protein sequences can have high pTM/pLDDT, all of them could be good candidates, and this promotes high diversity and low recovery. Consequently, when both CE and SC terms are present and when appropriate balance between them is found (in our case it is CE+SC, corresponding to the orange bar in both panels), we get a full benefit, i.e., the accurate recovery and high diversity of the generated protein sequences.

| | Model | Recovery | Recovery Change | Diversity | Diversity Change |
|---|---|---|---|---|---|
| 1 | GVP-GNN (1M) Jing et al. (2020) | 40.2 | – | NA | – |
| 2 | GVP-GNN (1M) Hsu et al. (2022) | 42.2 | – | NA | – |
| 3 | GVP-GNN (1M) + AlphaFold2 data Hsu et al. (2022) | 38.6 | -3.6 (-8.5%) | NA | – |
| 4 | GVP-GNN (1M) (our experiment) | 38.6 | – | 15.1 | – |
| 5 | GVP-GNN (1M) + SC (our experiment) | 39.6 | +1.0 (+2.6%) | 21.1 | +6.0(+39.7%) |
| 6 | GVP-GNN-large (21M) Hsu et al. (2022) | 39.2 | – | NA | – |
| 7 | GVP-GNN-large (21M) (our experiment) | 39.0 | – | 16.7 | – |
| 8 | GVP-GNN-large (21M) + SC (our experiment) | 40.1 | +1.1(+2.8%) | 19.3 | +2.6(+15.6%) |

Table 11: Comparison of amino acid recovery rate of protein sequences generated by GVP on the test split of CATH dataset. First row is the original result from GVP authors, rows 2 and 3 show the results from ESM authors, and row 4 shows the result from our experiments. A small difference between the values in first, second and forth rows can be attributed to some discrepancies in experimental settings as well as model initialization. We can see that a simple data augmentation baseline results in 3.6 (or 8.5%) drop of recovery relative to the unaugmented GVP (1M). On the other hand, the use of SC regularization leads to 1.0 (or 2.6%) gain in recovery, signaling the benefit of the proposed distillation approach. For the GVP-GNN-large (21M) model, shown in rows 6, 7 and 8 we were able to recover results closer to the published ones (39.0 vs their 39.2). And when SC is applied, we again see a boost in recovery (40.1 vs 39.0), and diversity (19.3 vs 16.7).

# H  ADDITIONAL PERFORMANCE COMPARISONS OF SC REGULARIZATION

## H.1  ESM-IF

In this Section we compare GVP-GNN (1M and 21M) performance under different training scenarios (CATH only and CATH + AlphaFold2 data) and present the results in Tables 11 and 12. The first row in Table 11 is the recovery rate of the original GVP-GNN (1M) model as reported in Jing et al. (2020). The following two rows (2 and 3) are the results presented in the work of Hsu et al. (2022) (ESM-IF). Their evaluation showed that the vanilla GVP achieved a slightly higher recovery rate of 42.2. GVP+AlphaFold2 represents the GVP trained on augmented dataset (CATH + AlphaFold2-generated structure/sequence pairs). Interestingly, this simple data augmentation baseline showed worse performance as compared to the original GVP, and the authors had to significantly increase GVP capacity (from 1M to 21M) to get any benefit from the data augmentation. Moreover, note that such a data augmentation idea can also serve as the baseline for our approach of AFDistill regularization, since AFDistill was trained on AlphaFold2-generated data and it can be thought of as a compressed representation of that data.

The rows (4 and 5) show our evaluation results of the vanilla GVP-GNN (1M), achieving slightly lower base recovery rate of 38.6, while this same GVP but trained with AFDistill regularization achieves a boost in recovery (39.6) and significant increase in the sequence diversity ( +39.7% as we showed in Fig. 7). On GVP-GNN-large (21M) model we were able to recover results closer to the published ones (39.0 vs their 39.2). And when SC is applied we again see a boost in recovery (40.1 vs 39.0), and diversity (19.3 vs 16.7).

Finally, in Table 12 we followed the setup of Hsu et al. (2022) and trained GVP-GNN-large (21M) on large dataset of CATH+AlphaFold2 (12M sequences) and evaluated on CATH test set. The second row in the table shows our that our experiment recovered results similar to the ones reported in Hsu et al. (2022), while in third row we present the SC-regularized training using our AFDistill model. Clearly, the sequence recovery was improved and even more so the diversity of the generated sequences went up from 13.8 to 18.5.

| | Model | Recovery | Recovery Change | Diversity | Diversity Change |
|---|---|---|---|---|---|
| 1 | GVP-GNN-large (21M) Hsu et al. (2022) | 50.8 | – | NA | – |
| 2 | GVP-GNN-large (21M) (our experiment) | 50.5 | – | 13.8 | – |
| 3 | GVP-GNN-large (21M) + SC (our experiment) | 50.9 | +0.4(+0.8%) | 18.5 | +4.7(+34.0%) |

Table 12: Comparison of amino acid recovery rate of protein sequences generated by GVP-GNN-large (21M) on the test split of CATH dataset, while trained on CATH + AlphaFold2 dataset (12M sequences). As observed before on experiment in GVP-GNN-large (21M) + SC when trained on CATH only, here when SC is applied, we see a minor boost in recovery (40.9 vs 50.5), and more significant increase in protein sequence diversity (13.8 vs 18.5).

Therefore, comparing data augmentation and model distillation for the task of protein design, we see that for the GVP models (1M and 21M), AFDistill offers a clear advantage, providing a modest boost in recovery, while significantly increasing diversity of the generated sequences. This improvement after applying SC regularization occurs because the baseline techniques, which rely on CE in training, primarily emphasize accurate sequence recovery, neglecting other protein sequences that can achieve the desired structure. SC regularization encourages the consideration of many relevant and diverse protein sequences with high pTM/pLDDT scores as strong candidate sequences. This results in a moderate improvement in recovery and a significantly larger boost in diversity. Moreover, the distillation overhead is amortized, as we train AFDistill once and use it in many downstream applications. The data augmentation would require additional computational cost in every downstream application.

## H.2 GRAPH TRANSFORMER

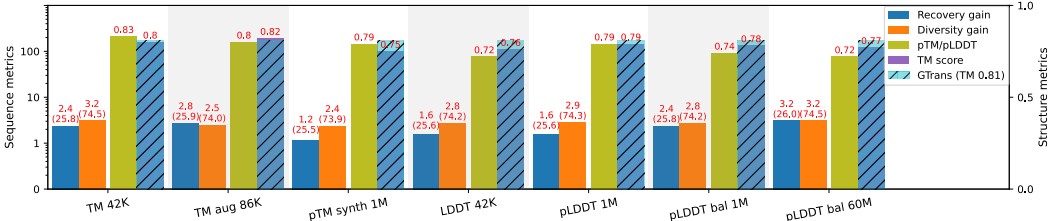

Figure 15: Evaluation results of Graph Transformer model trained with SC score regularization. Baseline model with no regularization achieves 25.2 in recovery, 72.2 in diversity and 0.81 in TM score on the test set.

We evaluated the effect of SC score on Graph Transformer (Wu et al., 2021), another inverse folding model, which seeks to improve standard GNNs to represent the protein 3D structure. Graph Transformer applies a permutation-invariant transformer module after GNN module to better represent the long-range pair-wise interactions between the graph nodes. The results of augmenting Graph Transformer training with SC score regularization are shown in Fig. 15 (see also Appendix, Table 14 for additional results). Baseline model with no regularization has 25.2 in recovery, 72.2 in diversity and 0.81 in TM score on the test set. As compared to GVP (Fig. 7), we can see that for this model, the recovery and diversity gains upon SC regularization are smaller. We also see that TM score of regularized model (TM 42K and TM augmented 86K pretraining) is slightly higher as compared to pLDDT-based models.

## H.3 PROTEIN INFILLING

Our proposed structure consistency regularization is quite general and not limited to the inverse folding task. Here we show its application on protein infilling task. Recall, that while the inverse folding task considers generating the entire protein sequence, conditioned on a given structure, infilling focuses on filling specific regions of a protein conditioned on a sequence/structure template.

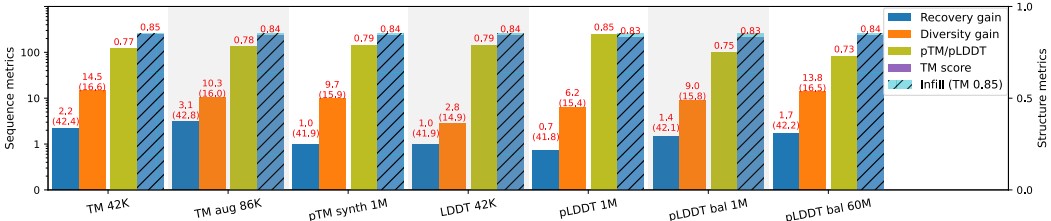

Figure 16: Evaluation results of Protein Infilling model trained with SC regularization. Baseline model achieves 41.5 in recovery, 14.5 in diversity and 0.80 in TM score on the test set. Similar as for the other applications, we see an improvement in the sequence recovery and even bigger gain in diversity. TM score shows that the resulting 3D structure remains close to the original, confirming the benefit of using SC score for training regularization.

The complementarity-determining regions (CDRs) of an antibody protein are of particular interest as they determine the antigen binding affinity and specificity. We follow the framework of (Jin et al., 2021) which formulates the problem as generation of the CDRs conditioned on a fixed framework region. We focus on CDR-H3 and use a baseline pretrained protein model ProtBERT (Elnaggar et al., 2020) finetuned on the infilling dataset, and use ProtBERT+SC as an alternative (finetuned with SC regularization). The CDR-H3 is masked and the objective is to reconstruct it using the rest of the protein sequence as a template. The results are shown in Fig. 16 (see also Appendix, Table 15 for additional results). Baseline model achieves 41.5 in recovery, 14.5 in diversity, and 0.80 in TM score on the test set. Similar as for the other applications, we see an improvement in the sequence recovery and even bigger gain in diversity, while using the AFDistill pretrained on TM 42K and TM augmented 86K, together with the pLDDT balanced datasets. TM score shows that the resulting 3D structure remains close to the original, confirming the benefit of using SC for training regularization.

## I  AFDISTILL EVALUATION ON DOWNSTREAM APPLICATIONS

Finally, in Tables 13 14, and 15 we show detailed results for GVP and Graph Transformer inverse folding task as well as protein infilling task. The table combines all the choices for AFDistill pretraining, showing their validation accuracy, and presents the corresponding performance on the downstream application without (top row in each table) and with SC regularization (all the following rows).

| Data | Distill model Training | | Validation | GVP Recovery | Diversity | Perplexity | pTM/ pLDDT | TM score |
|---|---|---|---|---|---|---|---|---|
| | Weighted sampling | Focal loss | CE loss | | | | | |
| – | – | – | – | 38.6 | 15.1 | 6.1 | 0.80 | 0.79 |
| . TM 42K | + | – | 1.37 | 36.8 | 22.2 | 6.3 | 0.78 | |
| | + | 1.0 | 1.16 | 37.6 | 21.1 | 6.0 | 0.87 | |
| | + | 3.0 | 1.10 | **39.6** | 21.1 | 5.9 | 0.84 | 0.84 |
| | + | 10.0 | 1.29 | 37.9 | 18.4 | 6.0 | 0.80 | |
| TM augmented 86K | – | – | 2.12 | 38.3 | 22.2 | 5.9 | 0.73 | |
| | + | 1.0 | 2.15 | **39.8** | 22.1 | 5.8 | 0.78 | 0.85 |
| | + | 3.0 | 2.19 | 37.8 | 19.8 | 6.1 | 0.73 | |
| | + | 10.0 | 2.25 | 38.5 | 21.2 | 5.9 | 0.72 | |
| TM synthetic 1M | – | – | 2.90 | 38.8 | 21.4 | 5.8 | 0.73 | |
| | + | 1.0 | 2.55 | **39.1** | 22.5 | 5.9 | 0.77 | 0.81 |
| | + | 3.0 | 2.75 | 39.0 | 21.9 | 5.8 | 0.74 | |
| | + | 10.0 | 3.20 | 39.0 | 22.0 | 5.9 | 0.69 | |
| LDDT 42K | – | – | 3.47 | **39.0** | 18.9 | 5.8 | 0.74 | 0.78 |
| | + | 1.0 | 3.44 | 38.7 | 22.5 | 5.8 | 0.73 | |
| | + | 3.0 | 3.42 | 38.9 | 21.2 | 5.8 | 0.73 | |
| | + | 10.0 | 3.39 | 38.5 | 22.3 | 5.9 | 0.72 | |
| . PLDDT 1M | – | – | 3.27 | **39.3** | 16.5 | 5.9 | 0.76 | 0.79 |
| | + | 1.0 | 3.28 | 38.8 | 15.5 | 5.9 | 0.72 | |
| | + | 3.0 | 3.25 | 38.9 | 18.2 | 5.8 | 0.78 | |
| | + | 10.0 | 3.24 | 38.4 | 16.2 | 6.0 | 0.73 | |
| LDDT chain 42K | – | – | 3.69 | 38.8 | 20.0 | 5.8 | 0.74 | |
| | + | 1.0 | 3.57 | **39.3** | 16.3 | 5.8 | 0.79 | 0.78 |
| | + | 3.0 | 3.63 | 38.9 | 15.9 | 5.9 | 0.72 | |
| | + | 10.0 | 3.59 | 37.9 | 23.2 | 6.0 | 0.73 | |
| pLDDT chain 1M | – | – | 3.29 | 39.4 | 17.4 | 5.8 | 0.78 | |
| | + | 1.0 | 3.36 | 38.7 | 16.3 | 5.8 | 0.76 | |
| | + | 3.0 | 3.30 | **39.6** | 18.3 | 5.7 | 0.79 | 0.77 |
| | + | 10.0 | 3.31 | 38.2 | 20.1 | 6.0 | 0.76 | |
| pLDDT balanced 1M | – | – | 2.63 | 39.1 | 17.1 | 5.8 | 0.75 | 0.82 |
| pLDDT balanced 10M | – | – | 2.43 | 39.3 | 17.7 | 5.9 | 0.73 | |
| pLDDT balanced 60M | – | – | 2.40 | **39.8** | 17.5 | 5.9 | 0.74 | 0.81 |
| pLDDT chain balanced 1M | – | – | 2.45 | 38.6 | 16.6 | 5.9 | 0.73 | |
| pLDDT chain balanced 10M | – | – | 2.24 | 39.1 | 17.8 | 5.8 | 0.73 | |
| pLDDT chain balanced 60M | – | – | 2.21 | **39.7** | 17.9 | 5.9 | 0.74 | 0.82 |

Table 13: Evaluation results of GVP inverse folding task, trained without (top row) and with SC regularization (all other rows). The table combines all the choices for AFDistill pretraining and showing their validation accuracy, as well as the corresponding performance on the downstream application. We select the best performance for each experiment based on the highest recovery rate (marked in bold).

| | Distill model | | | Graph Transformer | | | | |
|---|---|---|---|---|---|---|---|---|
| Data | Training | | Validation | Recovery | Diversity | Perplexity | pTM/ pLDDT | TM score |
| | Weighted sampling | Focal loss | CE loss | | | | | |
| – | – | – | – | 25.2 | 72.2 | 7.2 | 0.80 | 0.81 |
| TM 42K | + | – | 1.37 | 24.1 | 74.4 | 7.4 | 0.81 | |
| | + | 1.0 | 1.16 | 25.2 | 73.2 | 7.2 | 0.86 | |
| | + | 3.0 | 1.10 | **25.8** | 74.5 | 7.2 | 0.83 | 0.80 |
| | + | 10.0 | 1.29 | 24.9 | 73.9 | 7.3 | 0.81 | |
| TM augmented 86K | – | – | 2.12 | 25.0 | 73.3 | 7.1 | 0.78 | |
| | + | 1.0 | 2.15 | **25.9** | 74.0 | 7.1 | 0.80 | 0.82 |
| | + | 3.0 | 2.19 | 24.9 | 73.4 | 7.3 | 0.76 | |
| | + | 10.0 | 2.25 | 24.8 | 73.4 | 7.2 | 0.79 | |
| TM synthetic 1M | – | – | 2.90 | 25.3 | 73.2 | 7.1 | 0.72 | |
| | + | 1.0 | 2.55 | **25.5** | 73.9 | 7.2 | 0.79 | 0.75 |
| | + | 3.0 | 2.75 | 25.2 | 73.5 | 7.2 | 0.77 | |
| | + | 10.0 | 3.20 | 24.9 | 74.2 | 7.2 | 0.76 | |
| LDDT 42K | – | – | 3.47 | 25.4 | 73.2 | 7.1 | 0.75 | |
| | + | 1.0 | 3.44 | **25.7** | 74.2 | 7.1 | 0.72 | 0.76 |
| | + | 3.0 | 3.42 | 25.5 | 74.4 | 7.2 | 0.73 | |
| | + | 10.0 | 3.39 | 25.3 | 22.3 | 7.2 | 0.72 | |
| pLDDT 1M | – | – | 3.27 | 25.6 | 73.4 | 7.1 | 0.79 | |
| | + | 1.0 | 3.28 | 25.4 | 74.1 | 7.2 | 0.78 | |
| | + | 3.0 | 3.25 | **25.6** | 74.3 | 7.1 | 0.79 | 0.79 |
| | + | 10.0 | 3.24 | 25.4 | 74.0 | 7.1 | 0.77 | |
| LDDT chain 42K | – | – | 3.69 | 25.3 | 74.1 | 7.2 | 0.76 | |
| | + | 1.0 | 3.57 | **25.8** | 74.3 | 7.1 | 0.75 | 0.80 |
| | + | 3.0 | 3.63 | 25.5 | 74.2 | 7.1 | 0.77 | |
| | + | 10.0 | 3.59 | 25.6 | 74.1 | 7.2 | 0.76 | |
| pLDDT chain 1M | – | – | 3.29 | 25.3 | 74.3 | 7.1 | 0.78 | |
| | + | 1.0 | 3.36 | 25.2 | 74.1 | 7.1 | 0.76 | |
| | + | 3.0 | 3.30 | **25.6** | 74.4 | 7.2 | 0.79 | 0.81 |
| | + | 10.0 | 3.31 | 25.3 | 74.3 | 7.1 | 0.77 | |
| pLDDT balanced 1M | – | – | 2.63 | 25.8 | 74.2 | 7.1 | 0.70 | |
| pLDDT balanced 10M | – | – | 2.43 | 25.7 | 74.5 | 7.1 | 0.73 | |
| pLDDT balanced 60M | – | – | 2.40 | **26.0** | 74.2 | 7.2 | 0.74 | 0.78 |
| pLDDT chain balanced 1M | – | – | 2.45 | 25.7 | 74.3 | 7.1 | 0.72 | |
| pLDDT chain balanced 10M | – | – | 2.24 | 25.9 | 74.4 | 7.1 | 0.74 | |
| pLDDT chain balanced 60M | – | – | 2.21 | **25.9** | 74.5 | 7.2 | 0.72 | 0.77 |

Table 14: Evaluation results of Graph Transformer inverse folding task, trained without (top row) and with SC regularization (all other rows). The table combines all the choices for AFDistill pretraining and showing their validation accuracy, as well as the corresponding performance on the downstream application. We select the best performance for each experiment based on the highest recovery rate (marked in bold).

| | Distill model | | | CDR Infill | | | | |
|---|---|---|---|---|---|---|---|---|
| Data | Training | | Validation | Recovery | Diversity | Perplexity | pTM/ pLDDT | TM score |
| | Weighted sampling | Focal loss | CE loss | | | | | |
| – | – | – | – | 41.5 | 14.5 | 6.8 | 0.80 | 0.85 |
| TM 42K | + | – | 1.37 | 41.9 | 15.7 | 6.5 | 0.81 | |
| | + | 1.0 | 1.16 | **42.4** | 16.6 | 6.3 | 0.77 | 0.85 |
| | + | 3.0 | 1.10 | 41.7 | 14.6 | 6.7 | 0.78 | |
| | + | 10.0 | 1.29 | 40.8 | 14.4 | 6.6 | 0.79 | |
| TM augmented 86K | – | – | 2.12 | **42.8** | 15.5 | 6.5 | 0.78 | 0.84 |
| | + | 1.0 | 2.15 | 41.6 | 14.8 | 6.6 | 0.74 | |
| | + | 3.0 | 2.19 | 41.3 | 14.6 | 6.7 | 0.76 | |
| | + | 10.0 | 2.25 | 40.9 | 15.4 | 6.8 | 0.79 | |
| TM synthetic 1M | – | – | 2.90 | 41.8 | 16.0 | 6.6 | 0.79 | |
| | + | 1.0 | 2.55 | **41.9** | 15.9 | 6.7 | 0.79 | 0.84 |
| | + | 3.0 | 2.75 | 41.3 | 16.1 | 6.6 | 0.77 | |
| | + | 10.0 | 3.20 | 40.9 | 16.2 | 6.7 | 0.78 | |
| LDDT 42K | – | – | 3.47 | 41.3 | 15.1 | 6.5 | 0.83 | |
| | + | 1.0 | 3.44 | 40.3 | 15.5 | 6.7 | 0.84 | |
| | + | 3.0 | 3.42 | 40.8 | 14.4 | 6.8 | 0.81 | |
| | + | 10.0 | 3.39 | **41.9** | 14.9 | 6.6 | 0.79 | 0.84 |
| pLDDT 1M | – | – | 3.27 | **41.8** | 15.4 | 6.3 | 0.85 | 0.83 |
| | + | 1.0 | 3.28 | 40.7 | 14.3 | 6.5 | 0.85 | |
| | + | 3.0 | 3.25 | 41.7 | 17.2 | 6.5 | 0.84 | |
| | + | 10.0 | 3.24 | 41.6 | 16.1 | 6.6 | 0.85 | |
| LDDT chain 42K | – | – | 3.69 | 40.8 | 15.1 | 6.7 | 0.77 | |
| | + | 1.0 | 3.57 | 40.9 | 15.7 | 6.6 | 0.85 | |
| | + | 3.0 | 3.63 | **41.7** | 15.2 | 6.9 | 0.84 | 0.85 |
| | + | 10.0 | 3.59 | 41.6 | 15.2 | 6.8 | 0.83 | |
| pLDDT chain 1M | – | – | 3.29 | 40.5 | 16.1 | 6.6 | 0.81 | |
| | + | 1.0 | 3.36 | 40.8 | 17.1 | 6.5 | 0.88 | |
| | + | 3.0 | 3.30 | 41.0 | 15.0 | 6.5 | 0.85 | |
| | + | 10.0 | 3.31 | **41.8** | 15.4 | 6.3 | 0.87 | 0.85 |
| pLDDT balanced 1M | – | – | 2.63 | **42.1** | 15.8 | 6.4 | 0.75 | 0.83 |
| pLDDT balanced 10M | – | – | 2.43 | 42.0 | 14.9 | 7.0 | 0.76 | |
| pLDDT balanced 60M | – | – | 2.40 | 42.1 | 16.5 | 6.3 | 0.73 | |
| pLDDT chain balanced 1M | – | – | 2.45 | 41.1 | 18.0 | 6.1 | 0.75 | |
| pLDDT chain balanced 10M | – | – | 2.24 | 41.3 | 17.0 | 6.7 | 0.74 | |
| pLDDT chain balanced 60M | – | – | 2.21 | **41.9** | 17.5 | 6.3 | 0.73 | 0.83 |

Table 15: Evaluation results of Protein Infilling task, trained without (top row) and with SC regularization (all other rows). The table combines all the choices for AFDistill pretraining and showing their validation accuracy, as well as the corresponding performance on the downstream application. We select the best performance for each experiment based on the highest recovery rate (marked in bold).

