# OpenReview forum: "AlphaFold Distillation for Protein Design"
_ICLR.cc/2024/Conference — Submitted to ICLR 2024_

### Official Review · Reviewer_6HvL · 2023-10-27

**Soundness:** 1 poor
**Presentation:** 2 fair
**Contribution:** 2 fair
**Rating:** 3
**Confidence:** 5

**Summary:**

This paper proposes a method for inverse protein folding, which is the process of designing amino acid sequences that fold into a desired 3D structure. The key steps are:

- Knowledge distillation from AlphaFold to create a fast and differentiable model called AFDistill that predicts structural confidence scores (pTM, pLDDT) for a given sequence.

- Using AFDistill's predicted scores as a "structure consistency" (SC) regularization term when training inverse folding models like GVP, ProteinMPNN, etc.

**Strengths:**

S1. Novel idea of distilling AlphaFold into a fast and differentiable model (AFDistill) for structural consistency prediction.

S2. Elevation in recovery rate is observed though marginal.

**Weaknesses:**

W0. Given the marginal improvements, it is unconvincing that using plDDTs as loss (L_sc) for sequence design is really useful or not.

W1. The extra compute resources costs are significant in this plan, including both the distillation and the extra cost in evaluating / backpropagating L_sc. No justification of these extra computational costs is presented.

W2. A generated sequence with high plDDT generally means that it is *conservative* (easy to be predicted), but this is not an indicator that it corresponds to the target structure. Therefore, using plDDTs intuitively encourage "sequence stability" instead of "structural consistency". Two concepts are confused in this paper.

W3. Paper needs proofreading.

W4. In the evaluation part, the authors confused pTMs from AFD as TM. This metric lacks commonsense and is misleading, because it is totally irrelevant with the target structure. Under such evaluation, the elevation in so-called self consistency is trivial. The real TM between a structure prediction and the target structure should be reported.

**Questions:**

Q1. In AF2 the plDDT depends on the quality of MSAs and templates. How is the plDDT/pTM obtained? Especially for the augmented datasets where the authors run AF by themselves?

Q2. How would the authors justify that the smallest distillation set leads to largest improvements?

---

> ### Author Response · Authors · 2023-11-20
>
> Thank you for the reviews. We respond to your concerns below.
>
> **1. plDDTs as loss.**
>
> Yes, we found that plDDT as a loss is less effective as compared to pTM. This is why in most of the downstream applications we used the latter.
>
> **2.  Extra compute resources costs are significant.**
>
> That's not accurate. As we showed in Appendix in Tables 8 and 9:
>
> AFDistill training:
> - 24 hours on 1 GPU for TM 42K dataset
> - 7 days on 8 GPUs for pLDDT balanced 60M dataset
> - Cost is amortized: the model is trained once and reused it many downstream applications
>
> Downstream application training:
> - GVP (1M parameters) takes 24 hours on 1 GPU; GVP+SC takes 32 hours on 1GPU
> - PiFold takes 60 epochs (6 hours on 1GPU); PiFold+SC takes 60 epochs (8 hours on 1GPU)
> - Once the downstream application is trained, AFDistill is not used during inference.
>
> So, the additional cost is justified since we only train AFDistill once and use in all the downstream applications. Moreover, since AFDistill is not used during inference, the final inference time of the regularized model is the same as the original inverse folding model.
>
> **3. TM vs pTM**
>
> In our evaluations (e.g., see Figs. 7, 15, 16, Tables 13, 14, 15), we reported both predicted pTM value and the actual TM score. The point was to show consistency between these two values. This confirms our findings in Fig. 2, i.e., showing high correlation between AFDistill predictions of pTM/pLDDT and the actual values.
>
> **4. How is the plDDT/pTM**
>
> As we discussed in Section 3.1, we used AlfaFold Database to get the already precomputed pLDDT values for protein sequences. To get TM values, we computed TM score between AF2-predicted protein structures and the ground truth ones. For augmented data, we ran AF2 model for required protein sequences using the recommended settings in github.com/google-deepmind/alphafold.
>
> **5. Why the smallest distillation set leads to largest improvements?**
>
> The training set of TM augmented 86K lead to the best performing AFDistill model. This is likely due to the fact that we used true TM scores (between AF2-predicted and ground truth structures) for training, augmented with additional sequences to balance skewed TM value distribution. The other sets (e.g., pLDDT based) were less effective as they were based on AF2-self-predicted values, even though the corresponding datasets were significantly larger. Finally, adding synthetic pTM or pLDDT found to be also less effective.
>
> We hope that our responses clarify reviewer's questions and concerns and we hope they improve the assessment of our work.

---

> ### Comment · Reviewer_6HvL · 2023-11-21
> **reply to rebuttal**
>
> Thank you for the clarification. With regard to your rebuttal:
>
> 1) from the visualization, PTM and TM are far from being consistent to each other, especially when examing the plots in Figure 13. (R^2 may be a good metric in showing such "consistency".)
>
> 2) the improvements are too marginal to prove the distillation parts' contribution.
>
> Given the concerns above, I would not change the rating of the paper.

---

### Official Review · Reviewer_s5YM · 2023-10-29

**Soundness:** 2 fair
**Presentation:** 3 good
**Contribution:** 2 fair
**Rating:** 5
**Confidence:** 4

**Summary:**

In this work, the authors present AFDistill, a distilled model of AlphaFold2 that is used for protein inverse folding. AFDistill is based on ProtBERT and trained to predict the TM/LDDT and pTM/PLDDT of protein structures. The model is then used as a differential oracle for structure-consistency loss in training protein inverse folding model (e.g., GVP, ProteinMPNN, PiFold). The experiments show that the proposed model improves the diversity of predicted amino acid sequences and keep comparable performance in other metrics (recovery and perplexity).

**Strengths:**

1. The insight to distill AlphaFold2 for efficient inference in protein-related tasks is well-motivated.
2. Integration of structure consistency in inverse protein folding is important and should intuitively lead to better performance.

**Weaknesses:**

1. One major concern is the lack of technical novelty. AFDistill is based on existing ProtBERT without major modification of the model.
2. Though experimental results show significant gain in diversity of predicted amino acid sequences, the improvements on other metrics (e.g., recovery and perplexity) are trivial.

**Questions:**

Other questions besides Weakness:

1. In Fig. 1, there are lines (CE loss <-> AlphaFold) that cross with each other, which can make it confusing.
2. Why AFDistill uses discretized output (50 bins) instead of directly implementing regression tasks?
3. what is the value of $\alpha$ in Eq. 1 and how is it determined?
4. How does integration of AFDistill affect the training resources for protein inverse folding models?
5. The authors separate the experimental results of different models into different tables. It may be better to collect main results of different models (GVP, ProteinMPNN, PiFold, etc.) in one table for better illustration.
6. Different balanced datasets are introduced to train AFDistill. What is the main take-away of choosing which dataset for inverse folding?

---

> ### Author Response · Authors · 2023-11-20
>
> Thank you for recognizing the strengths of our work. We respond to your points and questions below.
>
> **1. Novelty**
>
> The novelty is in the direct estimation of pTM/pLDDT scores from a given protein sequence, bypassing protein structure estimation in AF2. The resulting AFDistill model has an interesting property that when used as a regularizer in protein design methods, this enables moderate improvement in sequence recovery and a significant gain in diversity. The ProtBERT model is simply a tool used in the construction of such a model (similarly as, e.g., current LLMs are used in many fine-tuning tasks to create novelty in the application stack). Therefore, our work should not be oversimplified while missing the novelty and practical benefit of AFDistill as a sequence-based pTM/pLDDT estimator and a structural consistency regularizer, whose benefits have been confirmed in 6 experimental results.
>
> **2. Metric improvement**
>
> For example, in Figure 7, when using TM aug 86K dataset for training AFDistill, the regularized GVP + SC gets 3.1% improvement in recovery and 44.1% gain in diversity over vanilla GVP. Even for recent state of the art approaches like ProteinMPNN or PiFold, we get around 1% increase in recovery and up to 15% boost in diversity. As another example, consider Table 11, where GVP-GNN-large + SC which gives 40.1 in sequence recovery (vs 39.0 for GVP-GNN-large) and 19.3 in diversity (vs 16.7 for GVP-GNN-large). This corresponds to 2.5% increase in recovery and 17.5% increase in diversity.
> The improvements in recovery are smaller as compared to diversity gains, but still not trivial, as claimed. Moreover, we believe that SC regularization is a novel and useful tool in the toolbox of a researcher that can improve baseline protein models through protein sequence regularization. With minimal overhead and clear gain in diversity while maintaining structural integrity.
>
> **3. Lines crossing in Fig.1**
>
>  Thank you, we'll fix it.
>
> **4. Why AFDistill uses classification rather than regression?**
>
> Classification is easier to handle and better performing using CE loss as compared to regression using least squares.
>
> **5. What is the value of alpha?**
>
> The alpha value used is 1, it was found through cross-validation.
>
> **6. Which dataset/model to use?**
>
> We have trained several models based on different datasets. Our evaluations point that AFDistill, trained on TM augmented 86K data, overall has a better performance as compared to others. Thus, it can be used as an initial approach for a given inverse folding problem. However, it should be noted that performance is problem-specific, and other models may need to be assessed as well.
>
> **7. How AFDistill affects training resources of protein inverse folding models?**
>
> - GVP (1M parameters) takes 24 hours on 1 GPU; GVP+SC takes 32 hours on 1GPU
> - PiFold takes 60 epochs (6 hours on 1GPU); PiFold+SC takes 60 epochs (8 hours on 1GPU)
> - Once the downstream application is trained, AFDistill is not used during inference.
>
> **8. Collect results into one table**
>
> Thank you for the suggestion!

---

### Official Review · Reviewer_XUpx · 2023-11-02

**Soundness:** 3 good
**Presentation:** 3 good
**Contribution:** 2 fair
**Rating:** 3
**Confidence:** 4

**Summary:**

The paper proposes AFDistill, a novel model that distills knowledge from AlphaFold to predict protein structural consistency (SC) scores pTM and pLDDT for a given sequence. AFDistill is used to regularize the training of inverse folding models by adding an SC loss term, which results in improved performance on benchmarks, boosting sequence recovery and diversity while maintaining structural integrity. Experiments demonstrate that SC regularization enhances inverse folding and infilling models, enabling accurate and diverse protein sequence generation. The fast, differentiable SC scores from AFDistill can also replace slower AlphaFold evaluations to cheaply assess structural properties of proteins.

**Strengths:**

- Utilize distillation method for transferring AlphaFold's knowledge into fast, differentiable SC scores.
- Implement AFDistill for cost-effective integration of AlphaFold expertise into design models.
- Conduct comprehensive experiments.

**Weaknesses:**

- I am unclear about the motivation behind this paper, particularly regarding the decision to utilize (distilled) AlphaFold instead of directly using AFDB. For instance, the paper states, "Despite this success, large-scale training is computationally expensive. A more efficient method could be to use a pre-trained forward folding model to guide the training of the inverse folding model." However, I fail to see the efficiency benefits of this approach, as utilizing the AF model (or distilled AF models) would entail additional on-the-fly inference costs compared to employing the AFDB.
- It is not clear whether the proposed method can outperform the model trained with AFDB or not.
- The overall performance improvement appears not much, and it is not clear where the gain comes from.

**Questions:**

- Is there a comparison between using AFDB and using AF-Distill?

---

> ### Author Response · Authors · 2023-11-20
>
> Thank you for taking time to read our paper and recognizing strengths of our work. We respond to your comments and questions below.
>
> **1. Utilizing the AF model (or distilled AF models) would entail additional on-the-fly inference costs compared to employing the AFDB**
>
> We would like to emphasize that our AFDistill regularization is only applied during training of inverse protein folding (see Fig.1. for a digram showing a clear difference between Training and Inference). The training time indeed is influenced but very minimally. Again, this comparison  was provided in Appendix, in Table 8 and 9. Let us provide this information here as well.
>
> In all experiments we used A100 GPUs.
>
> AFDistill training:
>
> - 24 hours on 1 GPU for TM 42K dataset
> - 7 days on 8 GPUs for pLDDT balanced 60M dataset
> - Cost is amortized: the model is trained once and reused it many downstream applications
>
> Downstream applications:
>
> - GVP (1M parameters) takes 24 hours on 1 GPU; GVP+SC takes 32 hours on 1GPU
> - PiFold takes 60 epochs (6 hours on 1GPU); PiFold+SC takes 60 epochs (8 hours on 1GPU)
> - Inflated training time is due to frequent validations (which involves sampling 100 samples per sequence for recovery and diversity computations).
>
> Once the downstream application is trained, AFDistill is not used during inference. So, the inference time is exactly the same as would be for the original inverse folding model, and also exactly the same as when using model with AF DB training.
>
> **2. Comparison of AFDistill-based and AFDB-based training**
>
> We use GVP example for illustration. Please refer to Table 11 and 12 in Appendix, and to Table 1 in Hsu et al. "Learning inverse folding from millions of predicted structures".
>
> From Table 1 of Hsu et al., you can see that for the original GVP-GNN (1M params) AlphaFold2 data augmentation showed worse performance as compare to a vanilla GVP. The authors had to significantly increase GVP capacity (from 1M to 21M) to get any benefit from data augmentation. On the other hand, in Table 11 of Appendix we showed that GVP + AFDistill trained on CATH alone achieves a boost in recovery and a significant increase in sequence diversity, showing a clear advantage over simple data augmentation.
>
> Similarly, for GVP-GNN-large (21M), trained on CATH, SC-regularization provides a 2.8% boost in recovery and 15.6% improvement in diversity over un-regularized training. Moving on to Table 12 of Appendix, for the model trained on CATH + AlphaFold2 (12M) we observed 0.8% increase in recovery and 34% increase in diversity.
>
> Please note that  it is inappropriate to compare GVP-GNN-large trained on AF2 data (12M) with GVP-GNN-large + SC trained on CATH data (21K). Even more incorrect is to expect that the latter can achieve similar performance as the former since AFDistill has no access to ground truth data and therefore 21K samples of CATH have less training power than 12M samples of AF2. On the other hand, comparing GVP-GNN-large trained on AF2 data with GVP-GNN-large trained on AF2 data and regularized with SC is meaningful and that is what our experiments are design for.
>
> **3. Is there a comparison between using AFDB and using AF-Distill?**
>
> Yes, there is a comparison in Tables 11 and 12. However, it's important to note that, for example, comparing GVP-GNN-large + AF2 to GVP-GNN-large + SC + CATH is not appropriate. It's unrealistic to expect GVP-GNN-large + SC + CATH to approach a 50.8% recovery rate, especially when it's exposed to only 21K CATH samples. Using SC alone doesn't lead to a significant boost in the recovery rate because it doesn't direct the system to a particular protein sequence that matches the input structure; it lacks access to the ground truth. However, SC regularization does encourage the consideration of many relevant and diverse protein sequences with high pTM/pLDDT scores as strong candidates. This results in some improvement in recovery and a significantly larger boost in diversity.
>
> To properly compare and evaluate the performance of the AFDistill-based SC: The impact of regularization should be assessed against the identical model and training data, but without the regularization. So, when a large GVP model is trained using 12M AF2 samples, the advantages of SC become evident when the same model and training data are used, but with the addition of AFDistill regularization. Training the large GVP model with SC regularization on a smaller CATH dataset is an inappropriate comparison and suggests a misinterpretation of our method.
>
> AFDistill-regularized training is not designed to replace training data, but rather to enhance the training with additional regularization guidance, which promotes recovery and diversity.

---

> > ### Author Response · Authors · 2023-11-20
> >
> > **4. The overall performance improvement appears not much, and it is not clear where the gain comes from.**
> >
> > Please check Figure 7. When using TM aug 86K dataset for training AFDistill, the regularized GVP + SC gets 3.1% improvement in recovery and 44.1% gain in diversity over vanilla GVP. Even for recent state of the art approaches like ProteinMPNN or PiFold, we get around 1% increase in recovery and up to 15% boost in diversity. Clearly, this is a significant improvement.
> >
> > The discussion of where the gains come from are given at the end of Section 8 of main paper and in Section G of Appendix. For instance, in Figure 14 as we increase the influence of SC regularization, the recovery rate decreases, and eventually, when CE contribution is very weak, we get low recovery and high diversity. As explained earlier, AFDistill has a limited guidance about the specific sequence to match a given 3D structure, because it lacks access to ground truth structure allowing many relevant sequences with high pTM/pLDDT to be considered good candidates. Consequently, when CE signal is too weak (where CE promotes recovery), the model mostly generates random sequences. On the other hand, when no SC signal is present, the model simply tries to memorize the data, achieving higher recovery rate but very small diversity. There is a sweet spot in between these two extremes (in Figure 14, this is CE + 1*SC) which balances the two forces, leading not only to large increase in diversity but also to an improvement in recovery over baseline CE training. In this regime, CE loss pulls the model towards ground truth, while SC offers other similar sequences which have high pTM/pLDDT scores, therefore likely matching the input structure better, and ensuring high recovery rate. At the same time these sequences are different from the ground truth, thus promoting diversity.
> >
> > *We hope that our responses address the comments made by the reviewer and we hope they improve the assessment of our work.*

---

> ### Comment · Reviewer_XUpx · 2023-11-22
> **Thank you for the rebuttal**
>
> 1. so the training cost of the proposed method is larger, and the cost of inference is the same. Why do you assert the proposed method is a "more efficient method"?
>
> 2. I think the motivation of this paper is to use AF-Distall to replace AFDB. Therefore, we should expect that "GVP-GNN-large + SC + CATH" would at least match, if not surpass, the performance of "GVP-GNN-large + AF2" (CATH + AFDB). Although you saying that "AFDistill has no access to ground truth data", AFDB data are no ground truths too. Both AFDistill and AFDB use the predicted results from alphafold2, the former is plddt/ptm and the latter is the predict structure.

---

### Meta-Review · Area_Chair_nkrm · 2023-12-15

**Metareview:**

The paper proposed to distill alphafold into a smaller more compute-friendly model as a regularization technique for improving inverse folding models for various applications. The method is relevant and clearly effective in increasing diversity of the designs but also to some extent in recovery of existing known sequences.


The reviewers are knowledgeable and are unanimously unconvinced by the relevance of the method and its performance. This is despite the thorough rebuttal that was offered by the authors which was attended to by the reviewers, though without leading to a change in their view.


While the AC sees the contributions of the method, they recommend rejection. The paper has to update its presentation of the motivation and empirical evidence for a next version especially when it comes to (1) computational burden and the rationale for accepting such burden, (2) the significance of the results both in diversity and recovery, (3) standing motivation despite (and possibly comparison to) models which are (or can be) trained on AFDB.

**Justification For Why Not Higher Score:**

The paper's presentation and empirical results are found unanimously unconvincing.

**Justification For Why Not Lower Score:**

N/A

---

### Decision · Program_Chairs · 2024-01-16

Reject